# In pursuit of a cure: The plural therapeutic landscape of onchocerciasis-associated epilepsy in Cameroon – A mixed methods study

**Maya Ronse**[1]*, **Julia Irani**[1], **Charlotte Gryseels**[1], **Tom Smekens**[1], **Serge Ekukole**[2], **Caroline Teh Monteh**[2], **Peter Tatah Ntaimah**[3], **Susan Dierickx**[4], **Kristien Verdonck**[1], **Robert Colebunders**[5], **Alfred K. Njamnshi**[2,6,7], **Sarah O'Neill**[1,8,9‡], **Koen Peeters Grietens**[1‡]

1 Department of Public Health, Institute of Tropical Medicine, Antwerp, Belgium, 2 Department of Internal Medicine (Neurology Unit), Faculty of Medicine and Biomedical Science, University of Yaoundé I, Yaoundé, Cameroon, 3 Department of Public Health and Hygiene, Faculty of Health Sciences, University of Buea, Buea, Cameroon, 4 Centre of Expertise on Gender, Diversity and Intersectionality (RHEA), Vrije Universiteit Brussel, Brussels, Belgium, 5 Global Health Institute, University of Antwerp, Antwerp, Belgium, 6 Department of Neurology, Central Hospital Yaoundé, Yaoundé, Cameroon, 7 Brain Research Africa Initiative (BRAIN), Yaoundé, Cameroon & Geneva, Switzerland, 8 Social Approaches to Health (CR5), Ecole de Santé Publique, Université Libre de Bruxelles, Brussels, Belgium, 9 Laboratoire Anthropologique des Mondes Contemporains (LAMC), Faculté de Philosophie et de Sciences Sociales, Université Libre de Bruxelles, Brussels, Belgium

‡ These authors share last authorship on this work.
* mronse@itg.be

## Abstract

### Background

A high prevalence of epilepsy has been observed in several onchocerciasis-endemic villages in the Sanaga River basin, Cameroon. Recent studies suggest that ivermectin, a drug that is distributed annually with the aim of eliminating onchocerciasis, may have a protective effect against acquiring onchocerciasis-associated epilepsy (OAE). This study, therefore, provides an in-depth understanding of both the complex therapeutic landscape for epilepsy as well as the experiences related to the 'community-directed treatment with ivermectin' (CDTI) campaign in order to identify a more trenchant path forward in the fight against epilepsy.

### Methodology/Principal findings

Based on a mixed methods study combining a qualitative strand with a quantitative survey, we found that epilepsy was perceived to have had an epidemic emergence in the past and was still considered an important health issue in the study area. Socio-economic status, availability and accessibility of drugs and practitioners, as well as perceived aetiology shaped therapeutic itineraries for epilepsy, which included frequenting (in)formal biomedical health care providers, indigenous and/or faith healing practitioners. Ivermectin uptake for onchocerciasis was generally well known and well regarded. The CDTI faced structural and logistical bottlenecks undermining equal access and optimal adherence to the drug.

**Data Availability Statement:** The data supporting the findings of this study/publication are retained at the Institute of Tropical Medicine, Antwerp and will

not be made openly accessible due to confidentiality concerns as the dataset cannot be fully anonymised given the nature of the research. Data can, however, be made available after approval of a motivated and written request to the Institute of Tropical Medicine at ITMresearchdataaccess@itg.be/.

**Funding:** This study was funded by ITM's SOFI programme supported by the Flemish Government, Science & Innovation (https://www. ewi-vlaanderen.be/en/department-economy-science-innovation) under subsidy number: EE145 4150 (SO, JI, MR, AKN, CTM, SE). The funders had no role in study design, data collection and analysis, decision to publish, or preparation of the manuscript.

**Competing interests:** The authors have declared that no competing interests exist.

## Conclusions/Significance

Locally accessible, uninterrupted, sustainable and comprehensive health-service delivery is essential to help alleviate the epilepsy burden on afflicted households. Addressing structural challenges of CDTI and communicating the potential link with epilepsy to local populations at risk could optimize the uptake of this potentially significant tool in OAE prevention.

## Author summary

Regions where onchocerciasis–a parasitical infection transmitted by blackflies–is endemic also tend to suffer from high levels of epilepsy. Recent studies suggest that ivermectin, an anti- onchocerciasis drug distributed annually to entire populations in onchocerciasis-endemic regions, may protect against developing onchocerciasis-associated epilepsy (OAE). As the link between onchocerciasis and epilepsy has been poorly understood and scientifically neglected in the past, our mixed methods research investigated how residents in an affected Cameroonian area perceive and cope with epilepsy; how they interpret the (causes of the) illness; where they seek care and why there. Armed with this knowledge, epilepsy control programs can optimize interventions geared at relieving the burden of epilepsy–and potentially OAE–which is essential given the fact that, despite 15–20 years of ivermectin distribution, onchocerciasis transmission persists and epilepsy prevalence in these regions remains high. Our findings illustrate how crucial it is to ensure locally accessible, uninterrupted, sustainable and comprehensive health service delivery for epilepsy. Furthermore, the structural challenges associated with the mass ivermectin distribution campaign must be addressed in order to relieve the burden of onchocerciasis, and potentially OAE. Without first addressing these structural bottlenecks, uptake and adherence to ivermectin treatment will remain insufficient.

## Introduction

In Cameroon, a high prevalence of epilepsy has been reported in several onchocerciasis-endemic villages in the Sanaga River basin (Centre and Littoral Region) [1–4]. Epilepsy is "*a brain disease characterized by abnormal brain activity causing seizures or unusual behaviour, sensations and sometimes loss of awareness*" ([5] p.xiii). Epilepsy can have a variety of aetiologies: genetic, infectious, metabolic, immunological, but the cause is also sometimes unknown [5]. It is considered to affect less than one percent of the global population, with a higher prevalence of epilepsy in low- and middle income countries (overall lifetime prevalence of 8.75 per 1000) compared to high-income countries (5.18 per 1000) [5]. Epilepsy is also characterized by the neurobiological, cognitive and psychological consequences of this condition. The condition also results in considerable psychological, social and economic ramifications for people living with epilepsy and their relatives [5–8]. For example, research has repeatedly shown that people with epilepsy can be confronted with stigmatisation and discrimination. They are more likely to abandon school at an early age and have reduced prospects of employment and marriage. There are also increased health risks including burning themselves and drowning due to unpredictable seizures. Females are more heavily affected than their male counterparts, with girls and women more likely to carry the burden of single motherhood and prone to experiences of sexual violence [5,7,8]. Treatment is symptomatic and not straightforward: a variety of antiseizure medicines exist with different indications, contraindications and potential side-

effects requiring continuous follow-up while wide price ranges and market-availability have to be considered within the (often) life-long span of the condition.

In onchocerciasis endemic areas, evidence has suggested that there is an association between onchocerciasis and epilepsy–onchocerciasis-associated epilepsy (OAE)–including the specific form of epilepsy referred to as Nodding Syndrome [3,7,9–23]. Onchocerciasis, also called 'river blindness', is classically known as an eye and skin disease caused by the parasitic filarial worm, *Onchocerca volvulus*, transmitted by repeated bites from infected blackflies breeding along fast-flowing rivers and streams [24].

In Cameroon, the National Epilepsy Control Programme provides low cost symptomatic treatment for patients in affected areas. Notably, despite the current provision of low cost anti-seizure medicines by certain public health services, a number of studies have shown people with epilepsy pursue various pathways for epilepsy treatment, often resorting to alternative treatment options such as traditional healing [25–30].

In the Sanaga River basin, a cohort study showed that *O. volvulus* microfilarial density in children in their early years was associated with the development of epilepsy later in life [3]. A recent study has shown that in these villages a high proportion of people suffering from a wide clinical spectrum of epilepsy met the criteria of OAE, defined by the authors as "(*i*) *being a PWE [person with epilepsy] according to the ILAE [International League Against Epilepsy] 2014 definition [see* [4]*]; (ii) age of onset of seizures between 3 and 18 years; (iii) no obvious cause for epilepsy from the history and physical examination; (iv) having lived in an onchocerciasis-endemic area for at least 3 years; (v) no history of abnormal psychomotor development prior to epilepsy onset*"([4] p72).

Still, the exact pathogenesis of OAE currently remains unknown and only symptomatic treatment is available. Mass drug administration (MDA) of ivermectin, through the 'community-directed treatment with ivermectin' (CDTI) campaign, has been on-going in several onchocerciasis-endemic areas in sub-Saharan Africa once or twice a year to eliminate *O. volvulus* microfilaria in the human population (by killing larvae living in human hosts). Between 1998 and 2015, the African Programme for Onchocerciasis Control and several non-governmental organizations provided support in the organisation of the campaign, with the aim of progressively transferring complete ownership to national authorities and local communities. The founding principle of the CDTI strategy is that the distribution of ivermectin relies on active participation of communities. The overall aim of this programme has been to eliminate onchocerciasis over a period of 15–20 years. Progress has been made, but elimination has not yet been achieved, with remaining foci of infection in Central and Eastern Africa, including in Cameroon [1,31–38].

The link between ivermectin uptake and development of OAE is currently under investigation with several recent studies suggesting that ivermectin may have a protective effect against the acquisition of OAE [3,7,36,39–41]. Notably however, recent epilepsy surveys in onchocerciasis-endemic villages in the Sanaga River basin of Cameroon showed that despite more than 20 years of annual MDA of ivermectin (Mectizan brand) through CDTI, high on-going onchocerciasis transmission and a high prevalence of epilepsy remains [1,36].

Our mixed methods study aimed to provide insights into the health seeking behaviour for epilepsy, as well as into the uptake of MDA with ivermectin through the on-going CDTI in the Sanaga River Basin, the latter of which might have an important role to play in the *prevention* of OAE.

## Methods

### Ethics statement

The study protocol was reviewed and approved by the Institutional Review Board of the Institute of Tropical Medicine in Antwerp (IRB/AB/ac/036 Ref: 983/14); the Ethics Committee of

the University Teaching Hospital in Antwerp (UZA) (15/26/277) and in Cameroon by the Comité National d'Ethique de la Recherche pour la Santé Humaine (2015/10/655/CE/ CNERSH/SP) as well as by the Ministry of Public Health, Division of Health Operations Research in Cameroon (D30-177/L/MINSANTE/SG/DROS/TMC). The researchers followed the Code of Ethics of the American Anthropological Association. In accordance with the approved study protocols, both for the qualitative and quantitative strand, permission to conduct the research was obtained by the village chiefs (or appointed representatives when absent) before the start of data collection and individual oral consent (or assent with parental consent for minors) was obtained from all participants. Oral consent was preferred for several reasons: interviewees were not going to be put at any risk of being harmed physically or psychologically; the act of signing one's name when providing information during an interview (whether for qualitative interviews or for the questionnaires of the survey) can potentially rouse mistrust; respondents can be disinclined to answer sensitive questions if they were required to sign a form prior to taking the questionnaire; due to the principle of 'respondent reactivity', referring to the effect of the formality of the informed consent procedure on the reliability of the responses. Oral consent was therefore expected to improve response reliability.

## Study design

This research made up part of a larger study on the aetiology Nodding Syndrome. We set up a sequential mixed methods study consisting of a qualitative ethnographic research strand followed by a quantitative survey research strand, described in standard notation as: [QUAL ➔ quan] [42]. The purpose of the qualitative study was to gain an in-depth understanding of the health seeking behaviour for epilepsy as well as perceptions and experiences related to onchocerciasis and the ivermectin distribution programme. The quantitative cross-sectional population-based survey aimed to quantify preliminary findings from the qualitative strand.

## Study setting

The study was carried out in the health districts of Bafia and Ntui, Centre Region of Cameroon and in the health district of Nyanon (for one village, Kelleng), Littoral Region. The main ethnicities in the Bafia and Ntui districts are Sanaga, Bafia and Yambassa. In the district of Nyanon, the main ethnicity is Bassa. All these villages are situated in regions with relatively close proximity to the Sanaga and Mbam rivers and their tributaries, where blackfly breading sites have been documented [31]. They have been classified as meso/hyper endemic areas for onchocerciasis in the country's 2006 Rapid Epidemiological Mapping of Onchocerciasis.

In these areas, cocoa has been increasingly cultivated by smallholders for whom this constitutes the main cash crop, with the Centre region being the biggest producer [43]. Other crops are mainly cultivated for seasonal subsistence agriculture with occasional surplus for selling such as corn, peanut, cassava, macabo, yam, plantain, egusi, okra, sweet potatoes, beans, leafy greens, several fruits, etc.

Many farmers migrate to the area for seasonal labour and nomadic pastoralists (Mbororo, a Fulani ethnic subgroup) also seasonally pass through the region with their cattle on transhumance. Aside from a large influx of migrants from within the area and from neighbouring districts, many migrants also come from northern regions (Northern Cameroon, Nigeria, Chad, Mali).

Living standards differ substantially between different socio-economic classes, depending largely on the ability to own land and having the resources to plant cocoa. However, the volatile cocoa prices imposed by buyers along with recent changes in climate that have led to unpredictable seasonal changes, severely impact crop harvest and consequently some farmers' subsistence, which also dictate socio-economic wellbeing.

## Village selection

Considering the absence of systematic epilepsy prevalence data, villages were purposefully selected based on the combination of (i) epilepsy prevalence estimates available in academic and grey literature or shared in personal communications with researchers who conducted previous research in the area (M. Boussinesq; J. Kamgno and A. Njamnshi); (ii) the registered number of suspected epilepsy cases at the health centre level in 2015; and, (iii) key informant/stakeholder insights arrived at during a preliminary phase of the study. The aim was to identify contrasting villages in terms of epilepsy burden to include variation and to allow for Qualitative Comparative Analysis (the results of which will be published elsewhere). In total, eleven villages were sampled for the qualitative study (Table 1; Fig 1).

For the cross-sectional population-based survey, we classified all 11 villages by hypothesized high (> 2.5%) or low (≤ 2.5%) epilepsy prevalence (based on a combination of the above-mentioned sources of information). Nine villages were ultimately included in the survey (Table 1), with one village having to be excluded due to geo-location (Kelleng), and another having served as the survey pre-testing site (Tobagne).

## Participant selection

**Qualitative study.**   For the qualitative component of the study, informants were recruited using a multiple purposive sampling approach, including theoretical, snowball and emergent sampling techniques [42], which aimed at the gradual selection of a maximum variation of social profiles and experiences (namely in gender, age, occupation, knowledge of/experience with epilepsy, socio-political status, locality, place of origin). The final sample included various types of informants: community members (farmers, persons with epilepsy, caretakers, teachers, village leaders, retired elderly, herders, ivermectin community distributors, shopkeepers, pastors); public and private health care workers (nurses, doctors); indigenous healers as well as epilepsy and onchocerciasis experts with experience in this setting.

**Quantitative study.**   As village housing lists were not available, participants were randomly recruited based on a modified World Health Organization's Expanded Programme on Immunization (EPI) sampling approach [45] to select the compounds (households) [46], which ensured spatial variation of the indicators. This process entailed the mapping of all households with the help of an individual appointed by the village leader prior to the start of the survey. In large villages, the village was divided into clusters and EPI sampling was applied in each cluster to ensure spatial spread of the sample. Within each household (defined as a group of people routinely sharing the same meal for dinner), two adults (one man, one woman

**Table 1.  Overview of study villages and hypothesized epilepsy prevalence.**

|  | Village | Health area, District |
|---|---|---|
| High-hypothesized epilepsy prevalence (>2.5%) | Bayomen | Kon-Yambetta, Bafia |
|  | Kananga | Balamba, Bafia |
|  | Nyamongo | Ngoro, Ntui |
|  | Bialanguena | Nyamanga II, Ntui |
|  | Badissa | Nyamanga II, Ntui |
|  | Kelleng | Kelleng, Nyanon |
| Low-hypothesized epilepsy prevalence (≤ 2.5%) | Tcheckos | Bokito, Bafia |
|  | Tchékané | Tchékané, Bafia |
|  | Ondouano | Ngoro, Ntui, |
|  | Yebekolo | Nyamanga II, Bafia |
|  | Tobagne | Bokito, Bafia |

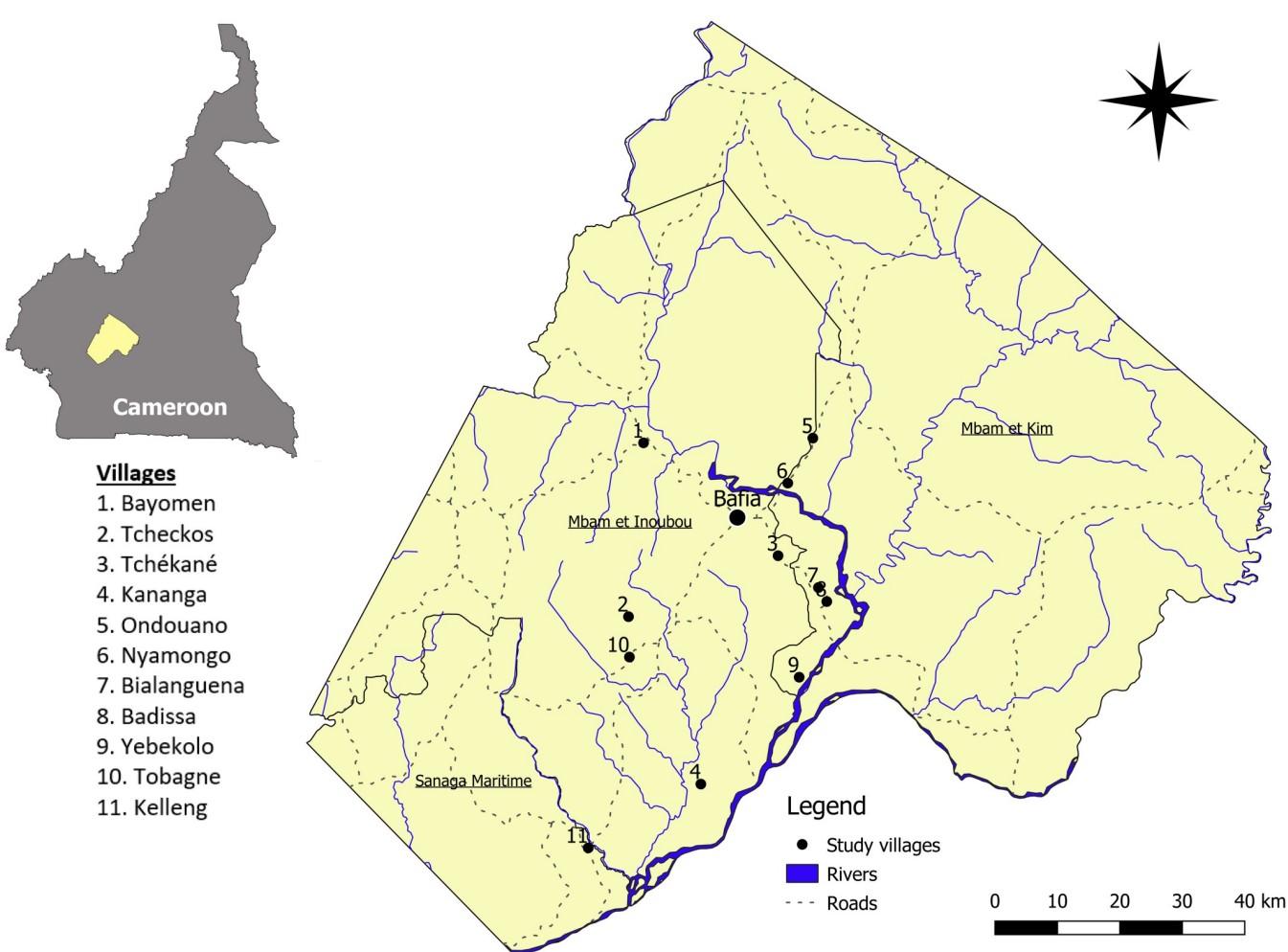

**Fig 1. Map of study villages in Cameroon (created by Rian Snijders using Quantum GIS 3.6.1 Las Noosa www.qgis.org) [44].**

being 16 years or older) were randomly selected and invited for participation in the study. In order to more actively involve respondents in the research process, the selection of participants was arrived at by listing and assigning numbers to all the men and women in a household (>16 years) and subsequently allowing a household member to randomly select the numbers. The surveying of men and women was done to ensure both genders' perspectives were represented in the data. The interviews were conducted at the same time, as far as was possible, in order to prevent one household member (male or female) from influencing the other's responses. For absences or refusals, other household members of the same gender were sought out and, if any were available and willing, asked to participate.

The sample size was calculated for the purpose of estimating proportions with 90% confidence intervals which have no more than a 10% margin of error. Since no information was available about either the population size or the expected prevalence of the indicators, we assumed a large population with proportions of 50% in order to get the largest sample size estimate. This resulted in 68 participants per village. To allow for estimating certain results for both women and men, we doubled the sample size (68 men, 68 women). We added a 10% buffer to account for missing data and non-response, therefore requiring 75 households (75*2 = 150 respondents) per village.

## Data collection

In both research strands, data collection was mainly carried out in French, which is widely spoken in the study area. When informants did not speak French, e.g. Pidgin, Fulfulde, Ewondo, research assistants and, in some cases, a member of the community translated.

**Qualitative data.** Data was intermittently collected by three researchers from the ITM (MR; JI; SO) with the help of two research assistants (SE; CTM) between February 2016 and May 2017, using ethnographic research techniques.

*Participant observation and informal conversations.* Participant observation was used to understand the local context and human practice in relation to the research questions. It was also vital to engender trust and confidence between participants and the research team especially when sensitive subjects were discussed, such as personal experience with epilepsy and consequent social exclusion and stigma. The researchers participated in daily activities in the community setting, while observing events in their usual context and having informal conversations with respondents. Informal conversations centred on emerging themes such as perceptions of epilepsy, access to and perceptions of public/private health facilities, traditional medicine and religious practices. This method was used to establish rapport, build trust with community members, reduce response bias and crosscheck the validity of the information obtained in the semi-structured interviews (see below). Notes were taken during informal interviews when appropriate; otherwise the conversations were summarised in detail after the conversation.

*In-depth interviews.* Semi-structured in-depth interviews were conducted in private or public places where participants felt at ease, such as at their residences or workspaces. Some interviews were performed at the health centre with patients waiting to consult with a neurologist. When appropriate, in-depth interviews were recorded and, when necessary, simultaneously translated to English by a trained field assistant. All interviews were later fully transcribed.

*Group discussions.* (Focus) group discussions were conducted with health professionals working on epilepsy and various community members. This method enriched the data by allowing different points of view to emerge in a discussion in which participants could build on what others said. These discussions were conducted at places where it is locally common to gather, such as a hospital office for health professionals, a gathering place at an epilepsy clinic for epilepsy patients and relatives, the residence of the village chief for community members, a school or a household's yard for household members. Depending on the context and type of topic covered, these discussions could either be formal or informal in nature. Notes were taken during the group discussions; some were recorded and all were later transcribed.

**Quantitative data.** Five trained interviewers using tablets with Open Data Kit (ODK) administered a face-to-face questionnaire. The questionnaire assessed participants' perceptions and experiences related to (i) epilepsy, (ii) ivermectin (distribution) and onchocerciasis; (iii) blackfly biting/nuisance and disease transmission, in addition to (iv) socio-demographic characteristics (S1 Appendix). This paper primarily focuses on variables related to (i), (ii) and (iv).

## Data analysis

**Qualitative data.** Data analysis for the qualitative study was carried out concurrently with data collection and formed an iterative process, where emerging results call for the adaptation of question guides and continuous re-testing of their validity in the field through triangulation of the different data collection techniques and types of informants. The preliminary analysis framed the design of the survey questionnaire. The analysis was retroductive (combining

inductive analysis from field data and theory from existing anthropological literature [47]). Special attention was paid to reflexivity, positionality and subjectivity throughout the ethnographic fieldwork and analysis, which was particularly insightful during periods of joint fieldwork by two researchers. NVivo 11 Qualitative Data Analysis software (QSR International Pty Ltd. Cardigan UK) was used to facilitate import, management and analysis of the data.

**Quantitative data.**   Quantitative data was analysed in R software using the survey package [48]. Descriptive statistics were used to characterize the study population and the main outcome variables for each village. Raw frequencies are presented in each analysis. Estimated proportions and associations were weighted according to the estimated total number of households in each village. The 90% confidence intervals for the estimates presented in the results section are provided in S1 Table. These generalize to the nine observed villages as a population and are constructed from a two-level cluster sampling design with variances adjusted for clustering in nine villages and 892 households. Weighting and finite population correction was applied for the households using the estimated total number of households in each village. During cleaning, some answer categories were added or renamed for analysis to include relevant additional information provided in the "other" open answer options. The closed-form questions of "don't know" and refusals were treated as valid responses and are part of the denominator in calculated proportions.

## Results

In this section, we describe the study participants, followed by an elaboration of the perceptions and experiences related to epilepsy, including perceived signs, prevalence and aetiology of the illness as well as health seeking behaviour in a plural therapeutic landscape. Finally, we describe our findings on the knowledge, experiences and perceptions of onchocerciasis and CDTI.

### Study participants

For the qualitative study, we conducted 94 in-depth interviews, 45 group discussions and 92 informal conversations. A total of 1313 people participated in the quantitative survey (Table 2). There were 44 instances in the survey where the sampled participant refused to participate or was not available.

**Socio-demographic characteristics of study participants.**   (S2 Table). The majority of participants in the survey were born in the region where the study took place (the Centre region), but not in the village where they were currently living showing a high rate of

**Table 2. Participants in the cross-sectional survey by village (quantitative strand).**

| Village | Total households in village | Households in study | Participants | Proportion of male participants |
|---|---|---|---|---|
| Bayomen | 282 | 75 | 124 | 50.8% |
| Kananga | 112 | 96 | 157 | 47.8% |
| Nyamongo | 251 | 104 | 166 | 50.6% |
| Bialanguena | 177 | 86 | 145 | 49.7% |
| Badissa | 140 | 113 | 184 | 47.8% |
| Tcheckos | 327 | 77 | 132 | 50.8% |
| Tchékané | 375 | 95 | 150 | 44.7% |
| Ondouano | 80 | 74 | 126 | 54.8% |
| Yebekolo | 340 | 80 | 129 | 51.2% |
| TOTAL | 2084 | 800 | 1313 | 49.3% |

migration to these villages and in this region in general (S1 Fig). These migrated participants had been living in the village for a median of 8 years (quartiles 3 and 20).

The main occupation was farming (75.3%; 1032). The seasonality of most income-generating crops, including cocoa, reportedly cultivated by 64.9% (793) of survey respondents, did not allow for stable livelihoods. While rarely being mentioned as a main occupation, fishing (27.0%; 431) and hunting (13.0%; 178) were additional subsistence activities for some. Most participants reported having had some form of education; 44.7% (629) attended primary school education and 37.1% (459) the first cycle of secondary school. The median age was 35 years (quartiles 27 and 50).

## Epilepsy

**Knowledge of epilepsy.** Epilepsy is locally known as the "falling-falling disease" (sometimes called "fainting sick" in Pidgin) or *la maladie de tombe-tombe*, referencing the typical tonic-clonic seizures of an epileptic patient. We classified participants' descriptions of epilepsy signs into four categories: (i) long term effects of epilepsy; (ii) signs during seizures; (iii) temporary signs immediately after seizures and (iv) precursor/trigger signs signalling upcoming seizures (Table 3). Survey respondents gave a median of 4 signs (quartiles 3 and 6), of which unexpected falling or collapsing (82.3%; 1092) and shaking (66.5%; 917) were most commonly given.

**Perceived prevalence of epilepsy.** The vast majority of survey respondents (94.3%; 1212) reported to know someone affected by epilepsy and 3.6% (64) reported having epilepsy themselves, with important variations between the villages; 20 participants reported not to know and 52 refused to answer the question (Table 4). Among those who self-reported having epilepsy, 71.1% (46) said it had been confirmed by a healthcare worker. Participants with self-reported epilepsy were on average 6.9 years younger than participants without it (raw mean ages are 31.5 versus 39.9). 3.3% (31) of men reported epilepsy versus 4.0% (33) of women (S1 Table).

**Table 3. Described signs of epilepsy in the cross-sectional survey (quantitative strand).**

| Effects of epilepsy (long-term) | Signs during seizures | Temporary signs immediately after seizures | Precursor signs of seizures |
|---|---|---|---|
| - scars from injuries<br>- arrogance and stubbornness<br>- mental problems<br>- behavioural change<br>- intellectual problems<br>- changes in physical strength<br>- changes in appetite<br>- changes in body weight | - collapsing on the floor<br>- partial or complete jerking of the body<br>- foaming of the mouth<br>- loss of awareness or consciousness<br>- stopping an action and becoming stiff<br>- screaming<br>- incontinence<br>- mouth injuring<br>- falling in the water or fire<br>- involuntarily chewing<br>- upward rolling of the eyes<br>- staring blankly/spacing out<br>- difficulties breathing<br>- self-inflicted harm<br>- dropping things held in hands<br>- repeated head nodding<br>- running off<br>- circling in one spot<br>- undressing oneself | - sleeping/snoring<br>- confusion and disorientation<br>- short-term memory loss | - missing a meal<br>- cold weather<br>- the (full) moon/night<br>- anxiety |

| Signs that could refer to several of the 4 categories |
|---|
| Communicative disruption/changes (e.g. speaking in a strange way); fatigue; "abnormal" behaviour; hallucinations (described as seeing things or talking to people who are not there); changes in eye colour or look/gaze; becoming calm/silent vs. becoming chatty and becoming aggressive/violent. |

**Table 4. Self-reported cases of epilepsy in the cross-sectional survey by village, gender and age (quantitative strand).**

| Village | Female participants who self-reported epilepsy (%) | Male participants who self-reported epilepsy (%) | Total participants who self-reported epilepsy (%) | Median age in years of participants who self-reported epilepsy (IQR) |
|---|---|---|---|---|
| Bayomen | 3 (5.6) | 0 (0) | 3 (2.7) | 45 (33.5–49) |
| Kananga | 6 (7.9) | 12 (16.4) | 18 (12.1) | 33 (25–36) |
| Nyamongo | 3 (4.0) | 1 (1.2) | 4 (2.5) | 24 (22–28.25) |
| Bialanguena | 7 (11.1) | 7 (10.1) | 14 (10.6) | 28.5 (22–33) |
| Badissa | 9 (9.6) | 6 (6.8) | 15 (8.2) | 31.5 (23–38.75) |
| Tcheckos | 1 (1.6) | 1 (1.5) | 2 (1.6) | 27.5 (27.25–27.75) |
| Tchékané | 1 (1.4) | 2 (3.1) | 3 (2.3) | 39 (34–45.5) |
| Ondouano | 1 (1.8) | 1 (1.5) | 2 (1.7) | 28 (26.5–29.5) |
| Yebekolo | 2 (3.3) | 1 (1.6) | 3 (2.4) | 36 (30.5–37) |
| TOTAL[a] | 33 (4.0) | 31 (3.3) | 64 (3.6) | 31.9 (23.0–41.6) |

[a] In this row, the percentages, the median and the IQR are weighted

Interviews during the qualitative study with elders additionally indicated that these high levels of epilepsy in the area were a relatively new phenomenon reported from the late seventies onwards. This is corroborated by survey results indicating that respondents who perceived the epilepsy problem as having worsened over time, were on average 11,1 years older (S1 Table) than those reporting that it has always been a problem (raw means were 37,1 and 48,6).

*"These matters [of epilepsy] started in . . . 78. It became worse from 78 onwards, the epidemic, it was as if it had exploded. So you would find four, five sick children in one same house".* (People affected by epilepsy and relatives, FGD)

During qualitative research, epilepsy was generally perceived to still be a prevailing problem. Informants often mentioned and observations showed that there are (still) households with several members affected by the disease. While observations and interviews with community members and health care providers indicated the continuing appearance of new cases, some respondents mentioned that there were currently fewer new/young cases than in the past. A few informants also expressed that individuals would hide their (or their relative's) epileptic condition out of fear of stigmatisation.

**Perceived aetiology of epilepsy.** The lack of (coherent) biomedical explanations for the high number of epilepsy cases in the villages made it difficult for people to make sense of this strange phenomenon and to trust health information from health workers during sensitisation messages or on the radio. When asked about the cause of epilepsy during the survey, 40.9% (570) of respondents cited not knowing the cause, while 5.6% (39) of these same respondents also cited at least one cause despite reporting being unaware of the origin of epilepsy (S1 Table). This paradox indicates the difficulty for participants to make sense of the illness. Both qualitative and quantitative strands showed that multiple aetiologies for epilepsy *co*-existed in this context, which can be summarized into two main types: (i) natural causes and (ii) mystical causes.

*Natural causes.* Epilepsy from 'natural' causes was thought to be a rare biomedical condition that had always existed and affected very few people. Respondents from both the qualitative and the quantitative strand described this type of epilepsy as potentially being linked to a genetic condition (being hereditary and/or a consequence of intermarriage between families); nerve/neurological problems; poor nutrition; being the consequence of another health issue (malaria, head trauma, complications during pregnancy or childbirth); blackfly (or other

insect) bites; worms or living close to water. Although the majority (75.8%; 997) of the survey participants thought epilepsy was not contagious, some respondents (from a combination of both strands) mentioned the belief that the illness could be transmitted by sharing food, sleeping next to or having sex with someone with epilepsy, through the foam coming out of a person's mouth during a seizure and through breastfeeding. The 'natural' kind of epilepsy was perceived to be more common prior to the sudden and unexplained perceived increase in people suffering from epileptic seizures, the latter of which most respondents attributed to sorcery.

*Mystical causes.* To the open question about the perceived cause of epilepsy, 46.0% (574) cited sorcery. When specifically asked whether epilepsy could be caused by sorcery, 84.0% (874) of respondents answered affirmatively (S1 Table). During qualitative research, sorcery-related epilepsy was often described as a new kind of epilepsy targeting mostly children and believed to be "cast at" or "injected" into people by sorcerers using mystical means. Informants from several villages explained that sorcerers buried cursed artefacts in schoolyards, at which point many children suddenly became epileptic. Epilepsy could also be "injected" into sweet fruits (e.g. mangoes, bananas, sugar cane. . .), infecting all who ate it. The malady could additionally be infused into mystical products purchased to generate wealth at home and on plantations, collecting a heavy toll by secretly hiding a spell cursing children in the household with epilepsy. This relatively recent development, perceived to affect mostly children and adolescents, was interpreted as an innovation in the field of sorcery, accounting for the sudden increase of epileptic cases.

> "Q: But if it's sorcery, why didn't it exist before? Because when we ask elder people, they tell us that 'no, when we were small, this didn't exist'.
>
> A: *You, yourself, know that technology advances every day. [. . .] It evolves in positive ways and in negative ways. [. . .] Even if it was here before [epilepsy caused by sorcery], it was at a manageable level. [. . .] And it wasn't within anyone's reach.*" (Mother of multiple affected children, farmer, IDI)

Informants explained that sorcerers, or people who consult sorcerers to inflict epilepsy on others, were motivated by jealousy, vengeance or greed for wealth and power. The transgression of social norms, such as stealing from someone else's crops, was also described as a trigger for the mystical affliction, adding a component of personal responsibility to the illness. Sorcerers were believed to be able to see children's 'stars' (i.e. destinies) and therefore, out of jealousy, would try to snuff out promising futures by inflicting this disease upon them. This can explain why some informants considered this epidemic-like illness to be destroying a whole generation.

> "Or you could be a baby child like this one here and the sorcerer has already seen your star. He knows that this child will be like this. So most often the cause is the star of the child and most often the cause is hatred, maybe from the family as such. 'Why is this family happy? Why are the children of this person like this? He has to be brought down". (Female farmer, relative of a traditional healer, IDI)

*Worms.* Both types of causation described above were believed to be able to interact. Worms directly embodied both the natural and the mystical level of causality. They were perceived to be able to enter the body and nest in the stomach and brain, causing epilepsy.

Worms were perceived to be a 'natural' element; yet they could also be inflicted upon you through sorcery.

> *"They imitate the epilepsies of the past. When somebody wants to make you epileptic, a worm must be cast at you. [..] We didn't know this before. In the past, when a child was born, he had the epilepsy of the past [the previous form of epilepsy]. Nowadays, when someone sees your baby like this, they cast epilepsy at it just like that".* (Elder women, farmer, IDI)

*Diet.* Informants also associated the consumption of specific foods with epilepsy, including slippery/slimy food such as fresh fish and okra/gombo, fresh/bloody meat from wild animals and chickens as well as sweet fruits. These foods were avoided by some, but not by all inhabitants, even if they were commonly believed to be associated with epilepsy. These foods served as an explanation for the frothy slimy saliva that sometimes accompanied a seizure. Smoked or dried fish, however, was not taboo. Specific villages had their own taboo animals, considered sacred (e.g. hippopotamus, turtles, crocodiles, wild cats) as they were said to have protected or saved the ancestors of the village in the past. The killing or consumption of these animals was also perceived as dangerous and a potential cause for morbidity or death.

**Therapeutic landscape.** The complexity of the illness interpretations within the local therapeutic landscape led to equally complex health seeking itineraries in the often-desperate quest for a cure. Epilepsy was generally said to be incurable, with only 'calming' the symptoms (i.e. reducing the number of seizures) as a way of managing the illness. Nevertheless, a few stories circulated of a complete cure after indigenous treatment or faith healing. Alternating and combining several types of care when searching for a diagnosis and treatment were common. Table 5 gives a summary of the different options, which are discussed below.

During the survey, respondents were asked where they would seek a solution for epilepsy in three different scenarios. Individuals who self-reported having or having had epilepsy were asked *(i)* where they sought treatment/a solution for themselves. Additionally, all participants were asked the hypothetical questions *(ii)* what their treatment choices would be in case a family member were to have epilepsy and *(iii)* in case epilepsy was caused by sorcery. In all 3

**Table 5. Therapeutic landscape for epilepsy in the study area.**

| Modality of care | Care provider | Care provision |
|---|---|---|
| Biomedical | Public health centres/hospitals | Clinical examination, diagnosis and prescription/ provision of antiseizure drugs, follow-up (sometimes) |
| | Private Catholic health centres/hospitals or mobile consultations by Sisters | |
| | Other private religious and lay (secular) health centres/hospitals | |
| | Private health centres/hospitals | |
| | Private home practitioners* | |
| | Pharmacies | Anti-epileptic drugs for self-treatment |
| | Shops/mobile drug sellers | |
| Indigenous | Seers | At (healer's) homes, using indigenous practices incl. applying food taboos, knowledge of herbs/barks |
| | Indigenous healers | |
| | Marabouts | |
| | Self-treatment | |
| Faith healing | Religious leaders, self-treatment | Prayers at church/home |

*Note: Some private home practitioners offer both biomedical and indigenous care

situations, a combination of several care-seeking channels was reported (median 2 for *(i)*, 2 for *(ii)* and 1 for *(iii)*). Biomedical channels were mentioned more often than not, in scenario *(i)* by 74.5% (49) and in scenario *(ii)* by 81.0% (1075), whether in combination with other channels or alone; however, not in scenario *(iii)* (46.9%; 659). To illustrate, S2 Fig provides an overview of the results for scenario *(ii)* and S3 Table a comparison of 'biomedical' vs. 'non-biomedical' treatment choices between 'low' and 'high' epilepsy prevalence villages in scenario *(ii)*.

**Biomedical care.** Biomedical care channels included public health centres and hospitals, offering epilepsy diagnosis and medication; Catholic health centres and hospitals, which are recognized private institutions providing mostly subsidised or free (when financially supported by the WHO) biomedical care for epilepsy patients and historically the first to provide epilepsy treatment in the area; other recognized private religious health structures; private home practices sometimes run by (retired) nurses from the public sector; the informal sector where pharmacies and (mobile) drug sellers provide over-the-counter epilepsy treatment.

**Indigenous care.** In contrast, when epilepsy was supposedly caused by sorcery–scenario *(iii)* in the survey–, traditional ('indigenous') healing was the most common survey answer (alone or in combination with other types of care). Three broad categories of indigenous healers exist that use herbs, barks, roots, placenta, rituals, incantations and spirit invocations, scarification and the promotion of dietary restrictions as treatments. First, diviners (*ngah*) were perceived to be in touch with the spiritual realm, guiding them in the divination of the cause/culprit of a health problem. Secondly, indigenous healers (*arondenah)* could treat with medicinal herbs and barks in combination with dietary restrictions after a diviner had determined the cause of the illness, and could be specialized in certain illnesses, including epilepsy. Third, *marabouts* had occult powers and worked through the incantation of spirits and use of rituals to either heal or inflict illness. In practice, indigenous healers could possess a combination of skills belonging to the three types described here. Additionally, self-treatment with herbal medicine was also described as a possible modality of care.

**Faith healing.** Some people with epilepsy turned to religious faith for a cure. A few branches of Protestantism, such as Pentecostal (e.g. Born-Again, Adventist) churches, treated through prayer and gave hope, but also promoted/required patients to exclusively put their faith in religion and abandon all other forms of treatment in order to leave the healing to God alone.

**Processes guiding therapeutic choice.**   Facing uncertainty about the aetiology of epilepsy, reliance on several therapeutic options in order to find relief and meaning was based on a process of aetiological interpretation, pursuit of an appropriate therapeutic option, evaluation of the treatment's effectiveness, re-evaluation of the possible aetiology based on treatment effectiveness, and adaption of the therapeutic itinerary if needed. During this process, individuals with epilepsy and their relatives also had to actively and pragmatically weigh the costs and benefits of various therapeutic options.

**Socio-economic status.** As described above, the dominant occupation in the study area was farming, where varying levels of socio-economic status were evident and where the seasonality of most income-generating crops did not foster socio-economic stability. Such a volatile economic situation undermined the ability to regularly purchase or access anti-epileptic treatment and medical follow-up. Farming as the primary occupation also signified that abandoning agricultural activities to seek treatment for a member would have direct repercussions on the income of and nutrition for the whole household. Among households with several members affected by epilepsy, the lack of financial means sometimes led to prioritising the purchase of anti-epileptic treatment for one affected member only, e.g. the member who was perceived to have seizures most frequently and/or for whom treatment was perceived to be most effective.

"Q: And why did she [her daughter] stop the treatment one year ago?

A: [. . .] To be frank, it's lack of means. [. . .] As I had already tried everything and it hadn't worked, and as I saw that for her, it [seizures] doesn't hit her regularly, I'm fighting to at least have the child's pills [her granddaughter]." (Mother with affected daughter and granddaughter, IDI)

**Perceived effectiveness.** When a biomedical treatment that was initiated after medical consultation or recommended by acquaintances, was perceived to be effective, the same medication and dosage was often continued for years without (re-)consulting medical practitioners. Other patients would only take/be given (by caretakers) anti-epileptic drugs during periods of seizures and not on a regular basis. However, when the regimen was perceived to fail, the whole biomedical channel was, in some cases, dropped from the therapeutic itinerary. If the cause was perceived to be mystical in nature, biomedical treatment could be used to help relieve the symptoms at any stage of the itinerary, although some informants mentioned that biomedical treatment could not help the patient until the curse itself was remedied through indigenous treatment channels. Intermittent periods of good health without seizures reinforced the perception that the adopted treatment, be it indigenous, faith or biomedical in nature, was effective.

**Drug shortages.** Drug shortages in public and private (catholic) health centres occurred occasionally and severely impacted people's health seeking behaviour, influencing the perceived benefits compared to other therapeutic options, selection of and adherence to biomedical care. When the drugs from the health centre were perceived to be effective in relieving symptoms, people felt compelled to go directly to a pharmacy (in distant cities sometimes) or (mobile) drug sellers in cases of drug shortages at the health centre and pay large sums of money for certain drugs that they would otherwise access for free or at a subsidised rate. Alternatively, some people would turn to indigenous or faith healing. Those who could not afford to purchase medicine from health centres on a regular basis or at higher prices elsewhere in case of drug shortages, bought treatment irregularly or remained at home without any treatment. Interrupting antiseizure medicines could lead to relapses as was mentioned by several health care workers and affected households, or even to the aggravation of symptoms according to neurologists working on epilepsy in Cameroon. The consequent aggravation of the disease hereby further threatened current and future individual but also household wellbeing.

**Distance and lack of infrastructure.** Depending on the village, public health centres could be far and road conditions did not always allow for easy access. Hence, individuals with epilepsy, or their relatives, did not always see the benefit of optimal adherence to long, sometimes expensive, anti-epileptic treatment regimens, the access to which could be both time-consuming and even more costly due to the long distances that needed to be covered. Patients were sometimes severely affected by the disease, which additionally complicated travelling outside of the village. This made it more practical often to buy treatment directly from street vendors or pharmacies.

**Patient follow-up.** Since epilepsy is a long-term (sometimes life-long) condition, and treatment is not linear (often necessitating treatment adaptations over time), patient follow-up is important. Health centres had different policies when it came to the follow-up of their patients. Catholic epilepsy clinics (health centres) tried to actively follow-up their patients by organizing monthly epilepsy consultation days at the clinic or mobile consultations in a central place. In some public health centres, individuals with epilepsy were passively followed-up (i.e. patients were supposed to come and ask for follow-up on their own initiative) while in others, where follow-up was intended to be more proactive, staff were challenged by the same obstacles

barring patients' access to formal biomedical health care, including lack of time, insufficient stock and a broad geographical coverage zone, especially when the health centre covered various villages with high epilepsy prevalence. Several health centre staff complained that patients did not make the effort of travelling to the health centre for follow-up visits unless they received the drugs for free or had a problem with the current treatment regimen and wanted a change. Some affected households also reported that they were chastised when they had missed a number of follow-up sessions and for this reason stopped attending entirely and opted instead to buy the drugs directly from vendors. In some cases, patients' relatives attended medical consultations or follow-up meetings instead of the patients themselves where the relative would then describe the patient's evolution in order to receive supplementary medicines.

## Onchocerciasis and community-directed treatment with ivermectin

**Knowledge about the aetiology of onchocerciasis and ivermectin.** In the onchocerciasis-endemic study region, annual MDA through CDTI of Mectizan (ivermectin) had been ongoing for the past 18 years (at the time of data collection) with the aim of eliminating onchocerciasis transmitted by blackflies (cf. introduction), and its associated complications, as part of the campaign "Integrated fight against onchocerciasis, lymphatic filariasis and intestinal worms".

*Blackflies and preventive measures.* The vast majority of respondents (88.5%; 1163) stated that blackflies can cause one or more disease(s) such as onchocerciasis/filaria(sis) (40.0%; 493), malaria (34.6; 364), skin rashes (31.4%; 381), skin diseases (15.3%; 190), eye problems (11.0%; 142), scabies (8.9%; 109). Epilepsy was mentioned by 1.6% (17). Local preventive measures mentioned to avoid blackfly biting consisted of covering their bodies (e.g. with long sleeves, socks, caps. . .) (86.1%; 1138), applying lime juice (11.0%; 146) or petrol (8.9%; 129) on the skin. Less frequently mentioned preventive methods included applying palm/kernel oil, making a fire, applying a mix of the previous mentioned products (e.g. palm oil + petrol), applying soap or taking Mectizan.

*Mectizan and perceived purposes.* The quantitative strand demonstrated that the vast majority of respondents had heard of Mectizan (98.5%; 1294) and were aware it was being distributed in their village (98.8% 1235) (S1 Table). Reported perceived purposes for Mectizan were: filaria (86.8%; 1135), skin rashes (71.8%; 951), (disease caused by) blackflies (51.7%; 718), eye (sight) problems (64.3%; 857), bumps/nodules (47.8%; 661), worms (42.3%; 559), onchocerciasis (39.8%; 535) and (disease caused by) mosquitoes (38.5%; 532). This confirms qualitative findings that terms like 'filaria' and 'skin rashes' were more commonly known than the term 'onchocerciasis'.

**Community-directed treatment with ivermectin.** *CDTI distribution.* Mectizan is distributed following the CDTI community-directed approach. In this way, the distribution of ivermectin is left in the hands of community leaders and community appointed distributors. Community distributors (CDs) are supposed to receive training each year right before the start of the distribution campaign. Except for external support during the training sessions, the CDs' work was not remunerated as it was considered to be the community's responsibility. Official guidelines for distributors of the drug state that all people living in an endemic community should receive ivermectin during a campaign, with the exception of (i) children under five years or measuring less than 90cm; (ii) severely ill people; (iii) pregnant women and (iv) women breastfeeding babies less than 8 days old. When asked how Mectizan was distributed to them the last time, the majority of survey participants answered that the drug was distributed to them at their house (62.6%; 710); 12.9% (237) went to the residence of the community distributor to get Mectizan. During qualitative research, these two methods were indeed widely

described and a 2-step-approach was also described in certain villages combining the two methods: a first phase where community members were asked to come retrieve the drug at a central place followed by a second phase where the distributors would go door-to-door distributing it to individuals missed in step 1.

*Uninterrupted long-term uptake.* A majority of respondents (61.6%; 745) in the survey reported to have spent periods of longer than one year without taking Mectizan since their first uptake (S1 Table) When comparing answers from villages classified as 'low epilepsy prevalence' with those classified as 'high epilepsy prevalence', there was a reliable difference between the two groups, with 56.5% (274) of people living in 'low prevalence' villages reporting *interrupted* uptake compared to 67.4% (471) in 'high prevalence' villages, although two villages in the 'high prevalence' group buck this trend (S4 Table). 18.6% (141) of (female) respondents stated they were pregnant or breastfeeding at the time of distribution therefore accounting for interrupted ivermectin uptake over the years.

Several bottlenecks were identified influencing distribution of, access to and uptake of mectizan

*Access.* In the survey, 24.9% (171) of respondents reported being away from the village as one reason for not being able to take Mectizan consistently on an annual basis. 20.0% (153) stated the distributors not having visited their house as a reason for non-uptake. Qualitative research revealed that migrants from other regions as well as individuals carrying out seasonal work, including non-resident workers, were sometimes purposefully excluded from distribution campaigns. However, workers residing in their fields during the harvesting period, which coincided with the distribution period, were also frequently missed. Other reasons for not taking ivermectin, based on a combination of qualitative and less-frequently-mentioned quantitative results, included: reports of favouritism and corruption in the distribution (e.g. CDs keeping the drugs and selling them for profit), people not being aware of the CDTI, the absence of a CD (e.g. CD having passed away or left the village without being replaced). Additionally, covering long distances and bad road conditions were barriers for CDs and inhabitants living far from the distribution point.

*Availability.* Stock-outs of Mectizan and/or gaps in regular (annual) distribution were reported barriers to the uptake of Mectizan in qualitative and (some answers from) quantitative data.

*Perceived effectiveness and side effects.* During the qualitative research, side effects were often mentioned when discussing Mectizan, such as swelling, itching and to a lesser extent, red eyes, fever, malaria symptoms and diarrhea. Nevertheless, 76.0% (984) of respondents still thought the benefits of taking Mectizan outweighed the potential side effects. Individuals widely stated that Mectizan was useful (87.7%; 1136) and improved the health situation in their village (68.6%; 845). Yet 18.4% (137) of survey respondents mentioned being afraid of side effects as a reason for not having taken ivermectin consistently on a yearly basis (S1 Table). The qualitative strand showed that some community members feared serious or even fatal side effects, with stories of individuals who had died after taking it in the past. Notably, several informants considered the (side) effects to have diminished over time, which to some meant a positive change because more reassuring to take, while to others translated into a decreased effectiveness of the distributed drug. Conversely, having the feeling that the drug had no effect or feeling otherwise healthy were other arguments stated by some respondents for not taking ivermectin in both strands.

*Distrust and motivation.* Based on qualitative data, distrust in the appointed CDs and/or village chiefs seemed to have deterred some community members from adhering to ivermectin uptake. Lack of motivation from CDs was additionally identified in some villages, especially since this work had to be done in addition to daily (work) activities. The distributor's work

was of voluntary nature for which communities were envisioned to find their own system of compensating such community work as part of the CDTI approach. Yet in some cases CDs' responsibilities seemed to be neglected, not valued or not trusted to be voluntary (i.e. not compensated) by certain community members.

*Ambiguity concerning distribution guidelines.* For CDs, community members and health care workers, certain common interpretations guided what constituted exclusion criteria for taking Mectizan. For instance, alcohol consumption prior to and/or after intake, failure to fast prior to intake, or being "too" old often barred individuals from receiving or taking Mectizan. Several respondents mentioned themselves not wanting to interrupt alcohol consumption as a reason for not taking Mectizan. Aside from the fact that these criteria were not stated in the official guidelines for CDs, varying interpretations of, for instance, what was considered being "too" ill, also revealed a high level of ambiguity concerning how to apply official criteria. Finally, ambiguity for CDs on how to cope with (unforeseen) absentees and newcomers (post population census), and how to systematically observe the administration of the drugs also proved problematic. Furthermore, CDs were sometimes faced with the dilemma of whether or not to provide individuals with Mectizan knowing that the drug would not be taken as prescribed; namely, some recipients preferred to apply Mectizan externally (mixed with body lotion applied to the skin).

## Discussion

This mixed methods study looked into the therapeutic landscape for epilepsy and community experiences with CDTI, as this is an important potential tool for the prevention of OAE. Our findings both illustrate the complexity of the health seeking itineraries of people afflicted with epilepsy and highlight that alleviating the epilepsy burden requires uninterrupted, sustainable and comprehensive health service delivery. The structural challenges of CDTI should be addressed to improve uptake and adherence, and communicating about the potential link between onchocerciasis and epilepsy could increase community motivation to participate in CDTI.

### Epilepsy

While epilepsy affects less than one per cent of the global population [5], our results show that epilepsy is considered an important and common health issue in the Sanaga River basin in Cameroon. The combination of the self-reported epilepsy numbers in our survey (4%) with the richness of the detailed epilepsy indicators described by community members (Table 3) substantiates this perception. These findings are consistent with previous reports of high epilepsy prevalence levels in the same area, particularly in villages located close to rivers [1,2,36,49,50]. Results from our qualitative research strand suggest continuous occurrence of new epilepsy cases to date. These findings correspond to recent epidemiological epilepsy studies conducted in the study area showing new cases of epilepsy in children and adolescents despite decreased incidence of epilepsy over the years [1,36].

The therapeutic landscape for epilepsy consists of a variety of treatment options, ranging from a variety of formal and informal, public and private biomedical (including for self-medication), indigenous and faith healing services. Treatment seeking itineraries are often quite complex with individuals alternating or combining several therapeutic options. The choice and sequence of these options is (re-)shaped by interrelated structural factors influencing the process of care seeking and adherence to treatment, including the socio-economic status of the afflicted household; availability of and accessibility to (formal and informal) biomedical health care, anti-epileptic drugs, indigenous and faith healing. Therapy choice is also, to a certain

extent, influenced by people's aetiological interpretations with people attributing epilepsy to natural causes, mystical causes, or a combination of both. These itineraries are, however, set to the backdrop of communities' acute awareness that formal biomedical knowledge currently provides neither a coherent aetiological explanation for (such high levels of) epilepsy nor a cure.

Mystical interpretations of epilepsy have to be understood within a context of deeply felt societal and structural inequalities. The frequent attribution of illness to sorcery–documented in various contexts throughout Sub-Saharan Africa [26,28,30,51–59]–should be interpreted as an attempt to make sense of inequalities in illness (burden), health and wealth.

Reflecting the findings from a study on Buruli Ulcer in Cameroon [59], our research indicates that perceived mystical causality does not exclude treatment seeking in the biomedical sector. Similarly, in line with Whyte's findings from a study on epilepsy in Tanzania, while people search for a treatment based on their aetiological interpretations, *"[n]ot only does interpretation affect choice of treatment, but outcome of treatment affects interpretation"* ([60] p.235). In this sense, if symptoms do not improve, potential causality is re-evaluated, followed by an exploration of other modalities of care–a process described as 'lay empiricism' [61]. As such, the health seeking itinerary continues to be re-shaped towards the objective of stabilised health in line with the household's means weighed against the direct, indirect and intangible health-related costs [7].

Therefore, while it is important to understand the different perceived illness aetiologies at the community-level, from a policy perspective, there are limits to the usefulness of distinguishing between sorcery and non-sorcery related illnesses. Both perceived aetiologies and associated therapeutic itineraries are based on pragmatic decisions related to structural barriers underlying everyday hardships, constituting a condition of structural violence, referring to longstanding unequal social (economic, political, historical, legal, religious, and cultural) structures that are in place in a society, obstructing the ability of individuals or populations to meet their fundamental human needs [62–64].

In this context, the main barriers to accessing effective relief from symptoms consist of (i) drug shortages in health centres, (ii) weak health infrastructures and (iii) non-comprehensive policies and clinical management of epilepsy [8,10,60,65], illustrated by the unsynchronized collaboration between public/private sectors catering to epilepsy patients. In addition, (iv) the lack of a coherent biomedical explanation for the disease and (v) known lack of a biomedical cure for epilepsy [60,66] do not inspire adherence to or trust in a straightforward biomedical trajectory. In such a context, mystical causes for epilepsy represent an arguably logical perception. This is particularly resonant with an illness that emerges uncontrollably at different points in life, with strange, poorly understood, sometimes long-term symptoms. Moreover, as the aforementioned factors lead patients to either fail to access, interrupt or altogether stop effective biomedical treatment–which people may pursue even if mystical causes are suspected–people are forced to seek alternative sources of relief which may include herbal treatments and other indigenous healing, faith healing and support offered by one of the religious movements in the area, or potentially substandard, degraded or falsified medication from an unregulated informal market.

## Onchocerciasis-associated epilepsy and community-directed treatment with ivermectin

Various community members described a sudden epidemic emergence of epilepsy in the past, of which no official documentation in the literature could be found. However, a high prevalence of epilepsy (around 8%) was noted by Boussinesq in the early nineties [2]. It should,

therefore, be investigated whether there was indeed such a sudden increase in the prevalence and incidence of epilepsy in the the Sanaga River basin before the nineties and what could have triggered such an increase. Although very few respondents mentioned a link between onchocerciasis (or the blackfly vector) and epilepsy, several studies are currently researching the possible causal relationship between onchocerciasis and OAE, and the potential protective effect of ivermectin on OAE [3,7,9–22,36,39–41]. Considering the potential link with onchocerciasis, possible explanations for the earlier epidemic-like occurrence of epilepsy in the region could include (a combination of) (i) a drastic increase in blackfly breeding sites due to the discontinuation of a period of subsidised residual indoor and outdoor insecticide spraying in the sixties and seventies (presumably as part of the *Service Général d'Hygiène Mobile et de Prophylaxie* campaign focusing on several endemic diseases such as sleeping sickness, onchocerciasis, leprosy and malaria); (ii) population growth in villages close to the rivers; (iii) increased farming activities–including cocoa farming–on the fertile soil along the borders of rivers leading to a closer proximity to blackfly breeding sites and increased shaded surfaces conducive to the breeding of blackflies; and/or (iv) the building of dams in the area (e.g. [67]), which has been shown, in some places, to increase blackfly breeding sites at the dam construction site and/or further downstream [68–71]. However, these remain hypotheses and further research is needed.

In this context where a perceived epidemic emergence of epilepsy occurred in the past, and where the region is wrought with social inequality, limited accessibility to and availability of effective therapeutic care, it is interesting to reflect on how the interpretations of epilepsy in this setting did not lead to illness interpretations with a political connotation, which was the case in Uganda for instance with regard to Nodding Syndrome (a type of OAE). In Uganda, the epidemiological peak of Nodding Syndrome coincided with internal conflicts and war during which population displacements played an important role in the interpretation of the illness [72–75]. However, in both contexts, the lack of a biomedical explanation for the increase in the disease made it difficult to make sense of the occurrence of the phenomenon which further triggered distrust in public health information provided at national and international level.

The proportion of actual epilepsy cases associated with onchocerciasis in the Sanaga River basin, remains unknown, even though recent studies have started filling this gap (see for instance [4,36] for Bayomen, Kelleng and Nyamongo villages). Were the scientific community to reach a consensus on such a link, and the positive impact of ivermectin uptake on preventing OAE, disseminating and integrating this knowledge in health information and communication in onchocerciasis-endemic villages, including in sensitisation campaigns for ivermectin MDA, could increase motivation to adhere to the CDTI programme and thus potentially increase ivermectin uptake.

Indeed, while ivermectin intake for onchocerciasis was overall well known and received in the study area, recent epilepsy surveys in onchocerciasis-endemic villages in the Sanaga River basin of Cameroon showed that there is still high on-going onchocerciasis transmission and a high prevalence of epilepsy, despite more than 20 years of annual MDA of ivermectin (Mectizan brand) through CDTI [1,36]. While this high prevalence of epilepsy may be due to other causes of the disease unrelated to onchocerciasis, the suboptimal uptake of ivermectin in the region, identified in our study, may also account for this high prevalence.

Our research determined that structural, logistical and motivational challenges with the CDTI implementation [76,77] undercut optimal uptake, such as a access barriers (to/from remote residents); reported stock-outs; unequal/discriminatory distribution of ivermectin; low adherence (due to feeling healthy, fear of side effects or perceived ineffectiveness of Mectizan, distrust in local actors involved in the distribution process); low motivation by CDs to

distribute Mectizan (work obligations, lack of support/recognition by the community of CDs' efforts); difficulty for CDs to observe drug administration and to address unforeseen scenarios (e.g. requested alternative use by recipients, residents temporarily absent, influx of seasonal migrants); and ambiguity surrounding CDTI exclusion criteria for CDs, community members and health care workers.

### Limitations

It should be noted that the classification of villages as 'high' or 'low' (epilepsy) prevalence has to be interpreted with caution as the classification could not always rely on the same level of secondary data. As described previously, the villages in the survey were purposively selected and do not allow for results to be generalized to a wider population, although the households in each village were randomly sampled. As such, the confidence intervals presented in S1 Table were constructed with these nine villages as a finite population. Proportions given in the text were weighted according to the villages' relative estimated population size, but frequencies were not. In addition, the numbers of self-reported epilepsy in the survey should be interpreted with caution since they were not clinically/neurologically confirmed.

### Conclusions

In a context where continuously accessible and high-quality medical care is simply non-existent, combined with local awareness of the absence of a biomedical cause or cure, believing in mystical aetiologies for the high occurrence of epilepsy is both pragmatic as well as negligible for disease control given the structural violence that dictates treatment choice. The structural roadblocks that impede regular access to quality treatment highlight the need for locally sustainable, uninterrupted, comprehensive and accessible health service delivery mechanisms, including the clinical management and follow-up of people with epilepsy and the availability of efficient/effective drugs. The current lack of these mechanisms is more detrimental to people's health than the existence of mystical aetiological interpretations, which, in this context, generally do not even exclude recourse to anti-epileptic drugs.

Were the scientific community to reach a consensus on the existence of a link between epilepsy and onchocerciasis in onchocerciasis-endemic areas and on the positive impact of ivermectin uptake on preventing OAE, disseminating this knowledge in affected villages could increase communities' motivation and adherence to the CDTI. However, while dissemination of knowledge about OAE would be essential for communities' understanding of the disease, failure to address the structural and logistic bottlenecks prevalent in the CDTI programme would signify continued sub-optimal uptake of ivermectin and the consequent persistence of onchocerciasis and OAE.

### Supporting information

**S1 Appendix. Survey questionnaire.**
(PDF)

**S1 Table. Confidence intervals for weighted proportions, means and medians presented in the text**[*]**.** For proportions, N = 1313.
(PDF)

**S2 Table. Characteristics (%) of the cross-sectional survey participants by village (quantitative strand).**
(PDF)

**S3 Table. Comparison of 'non-biomedical' treatment choices between 'low' and 'high' epilepsy prevalence villages in scenario *(ii)*.**
(PDF)

**S4 Table. Comparison of reported continuous ivermectin uptake between 'low' and 'high' epilepsy prevalence villages.**
(PDF)

**S1 Fig. Proportion of survey participants born in the study village.**
(TIF)

**S2 Fig. Set plot [78] showing the frequencies of treatment choices (Set Size) and of unique combinations of answers (Intersection Size) in case a family member were to have epilepsy.**
*Note*: The small graph on the left (Set Size) shows the individual frequencies for each answer category, while the graph on the right-side (Intersection Size) shows which unique combinations of answers were provided by the participants. As an illustration, the answer "hospital" was mentioned by 634 (49.1%) participants (cf. Set Size) and 159 (12.4%) participants mentioned it as the only treatment choice, while 475 (36.7%) mentioned "hospital" in combination with other options (cf. Intersection Size).
(TIF)

## Acknowledgments

We would like to thank everyone who gave us their time to talk with us; Adies Akyeiri Gbetie Bennin, Virgil Duparce Awoumou Ondhoua and Edmond N. Mouofo for their hard work in the field during the survey data collection; Rogers Nditanchou for his help during the first phase of the fieldwork; Rian Snijders for designing the map; Elizabeth Toomer for proofreading the paper, Patrick Mbonguele Isaac for his knowledge of the study region and guidance; as well as Adam Hendy, Joseph Kamgno, Michel Boussinesq and the other experts and researchers for the inspiring exchanges during the course of this work; and finally, we would like to thank the administrative and support colleagues.

## Author Contributions

**Conceptualization:** Maya Ronse, Julia Irani, Sarah O'Neill, Koen Peeters Grietens.

**Data curation:** Maya Ronse, Julia Irani.

**Formal analysis:** Maya Ronse, Julia Irani, Tom Smekens, Sarah O'Neill.

**Funding acquisition:** Sarah O'Neill, Koen Peeters Grietens.

**Investigation:** Maya Ronse, Julia Irani, Serge Ekukole, Caroline Teh Monteh, Sarah O'Neill.

**Methodology:** Maya Ronse, Tom Smekens, Kristien Verdonck, Sarah O'Neill, Koen Peeters Grietens.

**Project administration:** Sarah O'Neill.

**Software:** Maya Ronse, Julia Irani, Tom Smekens, Sarah O'Neill.

**Supervision:** Kristien Verdonck, Alfred K. Njamnshi, Sarah O'Neill, Koen Peeters Grietens.

**Validation:** Charlotte Gryseels, Peter Tatah Ntaimah, Robert Colebunders, Alfred K. Njamnshi, Sarah O'Neill, Koen Peeters Grietens.

**Visualization:** Tom Smekens.

**Writing – original draft:** Maya Ronse, Julia Irani, Charlotte Gryseels, Tom Smekens, Sarah O'Neill, Koen Peeters Grietens.

**Writing – review & editing:** Maya Ronse, Julia Irani, Charlotte Gryseels, Tom Smekens, Serge Ekukole, Caroline Teh Monteh, Peter Tatah Ntaimah, Susan Dierickx, Kristien Verdonck, Robert Colebunders, Alfred K. Njamnshi, Sarah O'Neill, Koen Peeters Grietens.

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
