## [Decision Letter · Decision Letter 0]

24 Oct 2020

Dear Mrs Ronse,

Thank you very much for submitting your manuscript "Epilepsy and onchocerciasis: the plural therapeutic landscape in pursuit of an epilepsy cure and experiences of ‘community-directed treatment with ivermectin’ in onchocerciasis-endemic Cameroon – a mixed methods study" for consideration at PLOS Neglected Tropical Diseases. As with all papers reviewed by the journal, your manuscript was reviewed by members of the editorial board and by several independent reviewers. In light of the reviews (below this email), we would like to invite the resubmission of a significantly-revised version that takes into account the reviewers' comments. 

We cannot make any decision about publication until we have seen the revised manuscript and your response to the reviewers' comments. Your revised manuscript is also likely to be sent to reviewers for further evaluation.

Sincerely,

Lisa Dikomitis

Associate Editor

Francesca Tamarozzi

Deputy Editor

Reviewer's Responses to Questions

**Key Review Criteria Required for Acceptance?**

**Methods**

-Are the objectives of the study clearly articulated with a clear testable hypothesis stated?

-Is the study design appropriate to address the stated objectives?

-Is the population clearly described and appropriate for the hypothesis being tested?

-Is the sample size sufficient to ensure adequate power to address the hypothesis being tested?

-Were correct statistical analysis used to support conclusions?

-Are there concerns about ethical or regulatory requirements being met?

Reviewer #1: The objectives of the study are clear, but the wording of the text makes it very difficult, if not impossible, to assess the appropriateness of the approach, methods, results and conclusions. This problem is common to the entire length of the text, so I will detail those problems together in "General Comments" section

Reviewer #2: The research objectives are clear, and the methods used are appropriate. No concerns regarding the statistical analysis or ethical issues.

Reviewer #3: Refer to attachment

**Results**

-Does the analysis presented match the analysis plan?

-Are the results clearly and completely presented?

-Are the figures (Tables, Images) of sufficient quality for clarity?

Reviewer #1: Results are not clearly presented. Tables are sufficient and clear.

Reviewer #2: Results are well presented in clear tables.

Reviewer #3: Refer to attachment

**Conclusions**

-Are the conclusions supported by the data presented?

-Are the limitations of analysis clearly described?

-Do the authors discuss how these data can be helpful to advance our understanding of the topic under study?

-Is public health relevance addressed?

Reviewer #1: Coclusions are mixed with methods and results. I miss some limitations, I'll explain in "General Commments"

Among other reasons, this study is very interesting because it analyzes together very different types of factors, which are influencing in the actual results of implemented health programs.

Reviewer #2: Conclusions are derived from the data. Discussions of the findings are relevant and shed some more light on the subject of onchocerciasis and epilepsy.

Reviewer #3: Refer to attachment

**Editorial and Data Presentation Modifications?**

Reviewer #1: Extension and structure are important concerns that should be modified.

Reviewer #2: I recommend acceptance after addressing a few minor comments.

Reviewer #3: Refer to attachment

**Summary and General Comments**

Reviewer #1: Authors, who demonstrate with so many and good bibliographic references of their own, that they are experts in the field, address an important health relevance problem.

Among other reasons, this study is very interesting because it analyzes together very different types of factors, which are influencing in the actual results of implemented health programs.

One concerning issue, in my opinion, is the Tittle. The use of “epilepsy cure” in the title is not adequate. It is misleading. Onchocerciasis related epilepsy, even, fight against epilepsy, would be better options

However, I think my most important concern is with the length and order of the writing structure. I believe that authors should consider making a drastic reduction in the length of the text, avoiding repetitions, phrases and paragraphs that do not add information, and data that, although they have other types of interest, are not necessary for the analysis that the study presents. Besides, I find the article would improve and would be much more attractive and comprehensive to the reader adding order in its structure. These problems have made it very difficult, if not impossible, for me to assess the appropriateness of the approach, methods, results, and conclusions. 

I can give some examples:

Regarding structure:

-ABSTRACT, Methodology/Principal findings: I would like a clear and separate summary of Methodology, before talking about results. Later in the text, the authors use some more clarifying phrases that could be used. Even, it’s not clear if they are presenting results or conclusions. 

-INTRODUCTION: Authors can consider to summarize, as background information, too many data that they present in Methods section. Those data are more explanations about why they decided to use those methods and make it difficult the understanding of the methods used. For instance, Methods are not the place to explain the background information: “Nodding Syndrome (a specific form of OAE)”. There is a lot of information in the Methods-Study Setting section, even in the results section (for example, the different types of traditional health providers explanation) that belongs to the background information. Besides, issues related to Methods are in results, as “For absences or refusals, other household members of the same gender were sought out and, if any were available and willing, asked to participate.”

It would be convenient to extract from the whole text what is background, what are methods used, what are results and what are conclusions, and try to write them in their corresponding section.

I needed to print out the text to understand under which heading some sub-headings were included. In spite of this, I was not able to understand the structure. In “375 Perceived aetiology of epilepsy”. It this a new epigraph? Should it be written in bolds as the previous epigraphs?, regarding “Natural” “Mistical” “Worms” and “Diet”, are they sub epigraphs of “Perceived aetiology of epilepsy”? I recommend making it clear with numbers or an organized structure of type of letters.

Same comments regarding “447 Therapeutic landscape”, “495 Processes guiding therapeutic choice” “587 Community-directed treatment with ivermectin”

Why are “287 Ethical considerations” in Methods? If not, please, made it clear. 

Regarding extension:

From lines 78-103, text could be considerably shortened. The article’s aim is not about epilepsy, but its relationship with Onchocerciasis. Authors should consider keeping just those data related with the topic of interest of this paper.

448-454: It would remove or considerably reduce some very complex sentences, and/or those that do not provide new information, because is already reflected in the text: “The complexity of the various illness interpretations within the vast and varied therapeutic landscape in our study setting led to health seeking itineraries that were equally complex, combining all manner of options in the often-desperate quest for a cure.” “Regardless of its perceived aetiology, epilepsy was generally said to be incurable, with only symptom suppression as a way of managing the illness. Nevertheless, a few stories circulated of a complete cure after indigenous treatment or faith healing. Alternating and combining several types of care in the process of searching for a diagnosis and treatment were common.”

397-495: It is very interesting to read about mystical causes and worms, with their anecdotes and literal stories included. But I consider that the extension is excessive and that it could be reduced without losing the understanding of the type of beliefs found. It is possible that these explanations could constitute another type of article, very interesting, but in this case it does not provide neccesary data to support the conclusions.

306-309. Introduction to results may be not necessary because the different data collected are already, or should be, in the methods.

For the entire text, I recommend analyzing the statements that are repeated several times in the different sections and avoiding them whenever possible.

I have another few questions for the authors: 

Regarding “Participant observation and informal conversations”: 

For the reasons that authors explain in the text, I find it a very interesting method that I, personally, had not known before. 

Perhaps the writing should reflect as weaknesses or strengths of the study a summary of the answers to the following questions:

-What validity, as data to analyze, do the authors give to these interviews? 

-Do they have some information about the reproductivity of the data obtained in this way? 

-Can they refer us to some studies in this respect? 

-How have they taken into account the subjectivity of the data obtained in this way? 

-Do the authors consider that they are the first ones to use this method to achieve conclusions? 

Ethics: Oral consent was preferred for different reasons. Can they explain the method used to register the actual given oral consent?

English language: I am not qualified to rate the correctness of language use.

Reviewer #2: Interesting piece of work which used both qualitative and quantitative research techniques to investigate epilepsy in onchocerciasis-endemic villages in Cameroon. the authors explored in-depth the different perceptions and therapeutic attitudes of people when it comes to epilepsy. Importantly, they highlight the barriers to a successful CDTI programme which is much needed to eliminate onchocerciasis and hopefully reduce the epilepsy burden.

I commend the efforts of the authors and recommend that this work be published after addressing a few minor concerns:

- Ref 39 in the manuscript is a conference abstract, and should be replaced by the full publication: Gumisiriza N, Kaiser C, Asaba G, et al. Changes in epilepsy burden after onchocerciasis elimination in a hyperendemic focus of western Uganda: a comparison of two population-based, cross-sectional studies. Lancet Infect Dis 2020; S1473309920301225. DOI: 10.1016/S1473-3099(20)30122-5

- On the subject of onchocerciasis control being able to reduce OAE burden, an important reference I would suggest to the authors is the meta-analysis by Siewe et al: Meta-analysis of epilepsy prevalence in West Africa and its relationship with onchocerciasis endemicity and control. Int Health 2020; 12: 192–202; DOI: 10.1093/inthealth/ihaa012. This has not been cited in the manuscript.

- Line 784, page 33: At the end of that line (... prevalence have to be...), please replace "have" with "has"

Reviewer #3: Refer to attachment

PLOS authors have the option to publish the peer review history of their article (what does this mean?). If published, this will include your full peer review and any attached files.

Reviewer #1: No

Reviewer #2: No

Reviewer #3: No
---

## [Decision Letter · Decision Letter 1]

4 Feb 2021

Dear Mrs Ronse,

We are pleased to inform you that your manuscript 'In pursuit of a cure: the plural therapeutic landscape of onchocerciasis-associated epilepsy in Cameroon – a mixed methods study' has been provisionally accepted for publication in PLOS Neglected Tropical Diseases.

Before your manuscript can be formally accepted you will need to complete some formatting changes, which you will receive in a follow up email. A member of our team will be in touch with a set of requests. In that changes, please make sure to include the change suggested by the reviewer: In the abstract (last sentence, line 56): Please replace the word "efficacy" with "uptake".

Best regards,

Lisa Dikomitis

Associate Editor

Francesca Tamarozzi

Deputy Editor

Reviewer's Responses to Questions

**Key Review Criteria Required for Acceptance?**

**Methods**

-Are the objectives of the study clearly articulated with a clear testable hypothesis stated?

-Is the study design appropriate to address the stated objectives?

-Is the population clearly described and appropriate for the hypothesis being tested?

-Is the sample size sufficient to ensure adequate power to address the hypothesis being tested?

-Were correct statistical analysis used to support conclusions?

-Are there concerns about ethical or regulatory requirements being met?

Reviewer #1: Methods are clear and appropriate

Reviewer #2: The objectives are well stated and appropriately tested. All ethical concerns have been adequately addressed.

**Results**

-Does the analysis presented match the analysis plan?

-Are the results clearly and completely presented?

-Are the figures (Tables, Images) of sufficient quality for clarity?

Reviewer #1: Yes

Reviewer #2: Data is well presented, both in the manuscript and as supplementary material. The data analysis that was reported reflects what was described in the methods.

**Conclusions**

-Are the conclusions supported by the data presented?

-Are the limitations of analysis clearly described?

-Do the authors discuss how these data can be helpful to advance our understanding of the topic under study?

-Is public health relevance addressed?

Reviewer #1: Yes

Reviewer #2: Conclusions stream directly from the study findings, with a proper discussion to put the current findings in perspective.

**Editorial and Data Presentation Modifications?**

Reviewer #1: No modifications needed

Reviewer #2: Accept

**Summary and General Comments**

Reviewer #1: The text has been substantially improved.

Reviewer #2: In the abstract (last sentence, line 56): Please replace the word "efficacy" with "uptake".

PLOS authors have the option to publish the peer review history of their article (what does this mean?). If published, this will include your full peer review and any attached files.

Reviewer #1: No

Reviewer #2: No

---

## [Editor Report · Acceptance letter]

18 Feb 2021

Dear Mrs Ronse,

We are delighted to inform you that your manuscript, "In pursuit of a cure: the plural therapeutic landscape of onchocerciasis-associated epilepsy in Cameroon – a mixed methods study," has been formally accepted for publication in PLOS Neglected Tropical Diseases.

Best regards,

Shaden Kamhawi

co-Editor-in-Chief

Paul Brindley

co-Editor-in-Chief
